# Functional Capacity of Noninstitutionalized Older Adults from Northwest Mexico: Reference Values

**DOI:** 10.3390/healthcare11121733

**Published:** 2023-06-13

**Authors:** Gabriel Núñez-Othón, Ena Monserrat Romero-Pérez, Néstor Antonio Camberos, Mario Alberto Horta-Gim, José Manuel Tánori-Tapia, José Antonio de Paz

**Affiliations:** 1Division of Biological Sciences and Health, University of Sonora, Hermosillo 83000, Mexico; gabriel@guaymas.uson.mx (G.N.-O.); nestorantonio.camberos@unison.mx (N.A.C.); mario.horta@unison.mx (M.A.H.-G.); josemanuel.tanori@unison.mx (J.M.T.-T.); japazf@unileon.es (J.A.d.P.); 2Institute of Biomedicine, University of León, 24071 León, Spain

**Keywords:** aging, physical performance, senior fitness test, decline muscle function

## Abstract

Introduction: Physical capacity (PC) is a strong determinant of health, quality of life, and functional independence in older adults. Having reference values for PC specific to a particular region allows for a contextual interpretation of an individual’s level. Objectives: The objectives of this study were to describe the evolution of key aspects of PC during the aging process and provide reference values for the major components of health-related PC for the older adult population in Northwest Mexico. Methods: A total of 550 independent older adults (60–84 years, 70% women) from the city of Hermosillo (Sonora, Mexico) were included between January and June 2019. PC was assessed using the Senior Fitness Test Battery (SFTB) and grip-strength test. Reference values were established for 5-year age groups, providing percentile values at 10, 25, 50, 75, and 90. The percentage decrease in functional capacity with aging was determined via a linear regression analysis of age against the percentage value of each subject relative to the average value of 60-year-old individuals of the same sex. Results: Statistically significant differences in the results between men and women within the same age group were few and inconsistent, except for handgrip strength, which was lower in women across all age groups. The functional level, with respect to reference values for each age and sex group, was similar between men and women. The most pronounced functional decline during the aging period occurs between 70 and 80 years of age. The various tests generally show an annual percentage loss of approximately 1% from 60 years of age. Conclusions: This is the first study in Mexico that provides reference values for physical capacity using the Senior Fitness Test Battery. In general, older adults—both men and women—show similar functional levels with respect to their respective reference values. In general, an annual decline of 1% from the age of 60 years occurs.

## 1. Introduction

The increase in the older adult population is a global phenomenon with important social, economic, and health consequences, directly impacting the increase in the demand for community social and health services [1].

In 2008, Mexico ranked fifteenth in the world in respect to the number of older adults as a percentage of its total population, with a value of 6.1%; by 2020, this proportion had reached 12.2%, which shows the accelerated rate of aging of its population [2]. This study was conducted in Hermosillo, the capital of the state of Sonora, in Northwestern Mexico, sharing a border to the north with the U.S. state of Arizona and to the west with the Sea of Cortes, being located in the Sonoran Desert; its climate is desert-like and extremely hot. According to data from the National Institute of Statistics and Geography (INEGI) in 2020, the estimated population of Hermosillo was approximately 968,000, and 12% of that population was over 60 years old [2]. In global terms, life expectancy has increased, and living conditions have improved in all populations. In Mexico, life expectancy at birth in 2019 was 76 years (78.9 for women and 73.1 for men), and healthy life expectancy was 65.8 years (67.2 years for women and 64.3 for men); these values are lower than the average for the WHO Region of the Americas [3].

The classic definition of functional physical capacity, as reported by Caspersen et al. (1985), is as follows: “the ability to perform daily tasks with vigor and alertness, without undue fatigue and with ample energy to enjoy leisure-time activities and to cope with unforeseen emergencies” [4]. The main health-related components of functional capacity are strength, aerobic endurance, flexibility, power, speed, agility, balance, and body composition [5,6]. An adequate functional physical capacity contributes to successful aging, and thus physical capacity assessment is also important in the context of the World Health Organization’s Global Strategy and Action Plan on Ageing and Health 2016–2020 as part of comprehensive care for older people [7].

Physical capacity can be measured in different ways, and the use of batteries consisting of different physical tests provides a global view of physical performance. The Senior Fitness Test (SCSRt) Battery is a group of tests that mainly assess the muscular strength of the lower and upper extremities, mobility, trunk and shoulder flexibility, and aerobic capacity [8].

Multiple studies have reported positive associations between higher levels of functional fitness and longer survival times [9], lower medication consumption, better mood [10], more years of independence [11], and higher quality of life [12].

The magnitude of each of the components of physical capacity progressively decreases with age and may differ by sex and race, and even by country or region within a country. These significant differences are explained by biological, social, and environmental factors [13]. Therefore, reference values are required in respect to age, sex, and country or region [14].

Contextualized reference values of the different components of physical fitness may allow the interpretation of the physical performance of an individual in comparison with their reference population and the identification of individuals in need of specific intervention [15]. As such, numerous publications provide normative values by age and country, in young people, in the general population, and in the aged population, for example, in Japan [16], Singapore [17], Norway [18], Colombia [19], Germany [20], Portugal [21], South Korea [22], Great Britain [23], and Canada [24]. However, normative values for physical fitness in the older Mexican population are not available.

The aims of this study were to describe the evolution of the main components of physical capacity throughout the aging process in older men and women in different age groups to provide reference values for the functional capacity of the older adult population of Northwestern Mexico.

## 2. Materials and Methods

### 2.1. Participants

This cross-sectional exploratory research included 550 independent older adults residing in the City of Hermosillo, Sonora, Mexico, and was carried out over a period of 6 months, from January 2019 to June 2019. Participants were recruited from municipality social clubs where noninstitutionalized older adults meet and socialize through leisure, cultural, and recreational activities. The inclusion criteria were those aged 60 years or older and those who had the ability to walk without assistance. The exclusion criteria were uncontrolled chronic diseases; medical contraindications for physical exertion; limitation of range of motion in the knee, hip, or lumbar spine that hinders proper and comfortable execution of the tests; institutionalization; or cognitive impairment detected via three or more errors in the Short Portable Mental Status Questionnaire [25].

The study was approved by the Ethics Committee of the University of Sonora (CEI-UNISON 12/2018) and registered in the Research Project Registration System of the University of Sonora (USO313003569). All participants signed an informed-consent form, and the study was conducted in compliance with the Declaration of Helsinki [26].

### 2.2. Procedure

Between 8 and 10 a.m. and after a period of two hours with no liquid or food, upon arrival at the laboratory, the participants remained seated in repose for 10 min. Their heart rate was then recorded with a pulse oximeter (Onyx II 9560, Nonin Medical Inc., Plymouth, MN, USA), and their blood pressure was recorded with an automatic device (Omron M6 IT Comfort^®^, HEM-7322U, Omron Healthcare, Kyoto, Japan) to verify that the participants were under clinical control at the time of the evaluation. Weight and height were measured (SECA, model 217, Hamburg, Germany), and body composition was assessed with a Tanita SC-331S Analyzer (Tanita Corporation, Tokyo, Japan).

#### Physical Assessments

After 5 to 10 min of a warm-up routine, the Senior Fitness Test Battery (SFTB) tests were performed in the order described, according to the protocol described by Rikli and Jones [8]: the chair stand test (CSt), in which the number of repetitions of getting up from and sitting down in a chair is counted; the arm curl test (ACt) for 30 s, in which the number of elbow bends performed while supporting a 5 lb dumbbell for women and an 8 lb dumbbell for men is determined; 8 ft Up&Go test (UGt), in which the time taken to rise from a chair and, without running, move 8 ft, turn back, and sit down is recorded; the chair sit-and-reach test (CSRt), in which the distance between the middle fingers of the hands and the tip of the toe when flexing the trunk forward while sitting in a chair with the knee of one limb extended and the ankle at 90 degrees is measured; the back-scratch test (BSt), which measures how close the middle fingers of the hands can be brought together behind the back by bending one elbow above and one below the shoulder; and the 2-minute step test (2St), which counts the number of times, in 2 min, that the right knee can reach the midpoint between the patella and iliac crest without moving from the starting zone.

An additional test (handgrip strength test (HGt)) was used; the dominant and nondominant hands were evaluated in an alternating order, each tested twice with a 30 s rest between the tests, with a hydraulic hand dynamometer (Jamar J00105, Bolingbrook, IL, USA). The subject performed the test in the standing position, with the elbow in full extension, and the upper extremity at a 90-degree internal rotation parallel to the major axis of the body [27], with the dynamometer handles positioned such that the individual felt comfortable in the grip [28]. For this test, the highest value of the four recordings was employed.

The percentile was calculated for the value obtained in each of the tests for each participant by comparing it with the age- and sex-specific normative values obtained from a population of older adults in the United States, as published by the creators of the Senior Fitness Test battery [8].

### 2.3. Statistical Analysis

The descriptive values of the quantitative variables are reported as the mean and standard deviation (SD). The normality of the distribution was verified with the Kolmogorov–Smirnov test with Lilliefors significance correction. Differences between the sexes were compared using Student’s *t*-test or the Mann–Whitney U test, depending on the normal distribution of the variables. Age groups of the same sex were compared by means of one-factor ANOVA.

The association between categorical variables was determined using the chi-squared test (χ^2^). The correlations between variables were determined with Pearson’s product-moment correlation coefficient.

For the statistical treatment of the different variables studied, outliers were excluded. For variables with a normal distribution, outliers were considered to be those above or below 3 standard deviations. In variables that were non-normally distributed, outliers were identified via the quartile (Q) and interquartile range (IQR), and outliers were considered to be those below Q1 − 1.5 IQR or above Q3 + 1.5 IQR.

To estimate the annual functional loss, we analyzed the linear relationship between age (x) and each of the functional variables (y) according to the general linear function “y = ax + b”, where a represents the slope, and b represents the intercept x in y. We analyzed the annual functional percentage loss with respect to the reference value considered as the mean of all subjects aged 60 years and separately for women and men aged 60 years.

To compare the annual percentage loss starting from 60 years of age among the different components of the assessed physical condition with that reported in other published studies that employed tests contained in the SFT on a similar age sample, we calculated the annual percentage loss of the data from those studies using linear regression analysis between age (or mean age within each age group) and the normative values published in those articles.

For each subject in each test, their functional level was determined in relation to reference values of individuals of the same age and gender, obtaining their percentile within the group based on data published by the creators of the SFT.

IBM SPSS 27.0 software (SPSS Inc., Chicago, IL, USA) was used for data analysis, and the statistical significance level was set at *p* < 0.05.

## 3. Results

A total of 550 functionally independent persons between 60 and 84 years of age, living in the city of Hermosillo (Sonora, Mexico), 70% of whom were women, participated in the study. The mean age was 69.4 ± 6.3 years for women and 69.5 ± 6.2 years for men.

Table 1 presents the anthropometric variables. In general, women weighed less than men and were shorter; however, significant differences were observed in BMI, in the 70–74 and 80–84 age groups, in which women had a higher BMI. Both men and women in the 80–84 age group tended to have a lower BMI than those in the younger groups. Although we found few significant differences between the sexes in BMI, we observed more significant differences in the percentage of body fat, which was higher in women.

Table 2 includes the number and proportion of men and women by age group with low, normal, overweight, or obese body weight based on the BMI cutoff points established by the WHO [29]. We noted a high prevalence of obesity or overweight. In total, 77.7% of the women in the sample were overweight or obese, as were 69.7% of the men. We found no association between age group and BMI in women (χ^2^ = 16.754; *p* = 0.159) or in men (χ^2^ = 16.227; *p* = 0.181).

The results of the functional tests that evaluated some of the manifestations of strength are presented in Table 3. Women had a lower HGt than men of the same age, whereas the result of the number of elbow flexions with a dumbbell in the hand was similar between men and women of the same age. According to the reference values published by Rikli and Jones [8], females of any age group had a higher percentile value in the ACt than males of the same age.

In the CSt test, women showed values similar to those of men of the same age, except for the 65–69 age group, and women in the 60–64 and 70–74 age groups showed higher percentile values in relation to the reference values for women compared to men.

Table 4 presents the results of the UGt and 2minSt. The Up&Go test time did not differ between the sexes for the same age group, except for the 65–69 group, in which women performed worse than men. In general, the two older groups showed significant differences compared with the younger groups. However, we found no differences among men and women from their reference groups in percentile values in any age group [8].

The results of the tests that evaluated flexibility and joint mobility are shown in Table 5. With the exception of the 80–84 age group, the women presented a wider range in the CSRt than the men, but without significant differences from the percentile values obtained by Rikli and Jones [8].

Women of all age groups had a wider range in the BSt than men of the same age. However, we found no significant differences between the sexes in the percentile value relative to their age and sex reference group.

In Figure 1, we graphically display the average percentile of the different age groups of the men and women in our sample, according to the reference values published by Rikli and Jones [8]. In general, the women in our sample presented a higher functional level in terms of the strength in their arms and legs than the men compared with their respective reference values.

Table 6 shows the values of the slope (loss or gain for each year), the intercept, and the index of determination of the linear regressions obtained for each of the functional variables analyzed, both globally for the entire sample and by sex.

Notably, the percentage loss of the different aspects of physical fitness was similar throughout aging, except for the loss of joint mobility, which was more accentuated. The error in the estimation of the loss of joint mobility was also large.

The values obtained from the evaluation of the different tests of the components of physical fitness that we performed are shown in Table 7, as reference values for each of the tests and for each of the age and sex groups, showing the values of the 10th, 25th, 50th, 75th, and 90th centiles.

## 4. Discussion

The values resulting from the assessment of different aspects of physical fitness throughout the aging process that we provide can serve as an initial approximation to be used as reference values obtained in a Mexican population. As expected, the loss of physical fitness components was generally progressive during the period of life known as old age, with some nuances, which we will discuss below. With each passing year, the loss is approximately 1% with respect to the reference value at 60 years of age.

Manual handgrip strength has been a widely monitored variable in epidemiological studies in which the study subjects are older adults, as the test is easy to perform and requires few material resources [30]. Positive correlations have been widely described between manual strength and survival rates and between cognitive function and levels of functional independence [31,32,33,34]. The women in this study showed significantly weaker handgrip strength than the men, as has been consistently found in publications that have assessed this variable in samples composed of men and women in both young adult and older adult populations [23,35,36,37]. From the age of 70 years is when the loss of manual prehensile strength starts to be substantial with respect to the values of the 60-year-old subjects in both men and women; one could hypothesize that specific grip=strength training during the period prior to the age of 70 may be of particular importance in reducing the rate of grip-strength decline at this age. However, further studies are needed to confirm or refute this hypothesis.

Based on the data from our study, starting at the age of 60, women in our sample experienced an annual grip-strength loss of 1.14%, while men exhibited an annual loss of 1.36%. This loss in women is lower compared to the reported loss in other studies of older women, showing losses of −1.41 [23], −1.3 [36], −1.38 [38], −1.95 [39], −1.6 [37], or −1.28 [22], and similar to −1.12 [14].

The percentage loss of grip strength in men from our sample is lower compared to the loss observed in other studies of older men, such as −1.41 [23], −1.95 [39], −1.44 [37], −1.44 [14], and similar to −1.38 [36], or higher than −1.26 [22], −1.26 [38].Grip strength is a variable used in sarcopenia screening, establishing cutoff points below which sarcopenia is suspected [40]. The European Sarcopenia Study Group uses hand-grip cutoff values of <16 kg for women and <27 kg for men [41]. According to this criterion, 7.2% of the women in our study and 14.5% of the men had handgrip strength below these cutoff points, and of those below these values, 80% of the women and 25% of the men were 80 years of age or older.

For the implementation of activities of daily living, in addition to manual grip strength [42], limb strength plays an important role [43]. The reason for including the assessment of upper-extremity strength, in addition to the handgrip test, in the overall evaluation of physical fitness in older individuals lies in the need to comprehensively evaluate their physical capabilities; in the SFT battery, the strength of elbow flexion is assessed using the arm-curl test (ACt) [8]. However, the results of this test for men and women are not directly comparable due to the difference in the weight of the dumbbell used, as indicated in the Methodology section. Furthermore, comparing the results of this test with data published by other authors is challenging. Although the SFT recommends a weight of 5 pounds (2.27 kg) for women and 8 pounds (3.63 kg) for men, slight modifications in dumbbell weight are often made in different published studies. This variability in dumbbell weight can potentially influence the test results, as some studies use 5- and 8-pound dumbbells [13,21], while others employ dumbbells weighing 2 and 3 kg [44]. Additionally, in some publications, the specific weight of the dumbbells used is not specified [45]. The reference values for this test, as obtained by Rikli and Jones [8], were obtained using dumbbells weighing 5 and 8 pounds, similar to our study. Using this reference and comparing their respective sexes, the women in our sample exhibit superior elbow flexion strength compared to men. In comparison to their respective age groups, women are approximately at the 80th percentile, while men are around the 55th percentile.

The annual percentage loss of this component of physical fitness, the number of elbow flexions against resistance, from the age of 60 years onward, was 0.68% in women and 0.76% in men. The disparity in the use of dumbbell weights complicates the comparison between age groups in different studies, but it has less impact on comparing the percentage loss since each study consistently uses the same dumbbell weight. The women in our sample show a similar percentage loss in this test compared to older women in studies, such as −0.68 [44] and −0.72 [46], and they exhibit lower losses than those reported in other studies, such as −1.02 [47], −2.04 [21], and −1.04 [45]. The annual percentage loss in men from our sample is lower compared to other studies, such as −1.02 [44], −1.16 [47], −1.62 [21], and −1.17 [13], but higher than what was observed in another study (−0.55, [45]).

Lower-limb strength is also an important predictive factor for functional independence in older adults [48], as it is an activity that is performed a considerable number of times per day (in mature adults, between 46 and 60 times per day [49]), and it often serves as a precursor to other actions, such as walking. Therefore, the chair stand test is likely one of the most commonly used tests in the assessment of older adults, both in clinical settings and research. There are numerous variations in its evaluation, such as the number of cycles in 5, 10, or 30 s, with eyes open or closed and on firm ground or foam [49]. The SFT battery proposes counting the number of cycles in 30 s (CSt), which is considered an indirect evaluation of knee-extension strength [50,51]. In older adults, the number of cycles is estimated to have a good correlation with leg press strength adjusted for body weight (r = 0.78 in women and 0.71 in men) [52]. Overall, the women in our sample (except for those in the 64–69 age group) did not show differences in performance on this test compared to men. The reason is that while men demonstrate greater absolute strength than women, women have a lower body weight, so sitting and standing up becomes an expression of relative strength (manifested strength/body mass). In relation to the reference values used [8], women demonstrate a better functional level in this test compared to men until the age of 80, being around the 70th percentile, while men are around the 56th percentile.

The percentage loss in CSt of women from our sample is generally higher compared to the loss observed in other studies on older women, such as −0.88 [47], −0.7 [44], −1.01 [15], −0.90 [46], or −1.43 [45], and only lower than one study that reported an annual percentage loss of −2.44 [21]. In CSt, the men from our sample also exhibit higher annual percentage losses compared to other studies on older adult males, such as −0.98 [47], −0.71 [44], −0.66 [13], or −1.04 [45]. They show a similar percentage loss to another study (−1.38, [15]) and are lower than one study (−2.09, [21]).

The Up&Go test, also known as the Timed Up and Go test, is frequently used in studies conducted with older people because of its statistical association with the risk of falls [52,53], disability [52], functional independence [54], and frailty [55]. The Up&Go test does not measure a single component of physical fitness in isolation, so it is employed as a measure of agility or ambulation ability [56], or as an indirect measure of balance [57,58].

In general, there were no differences in performance for the Up&Go test between men and women in our study, except for the 65–69 age group. Similarly, there were no significant differences in performance between different age groups of the same sex. However, as individuals age, there is an increase in the time required to complete the Up&Go test, indicating a decline in displacement speed. This decline is particularly noticeable during the transition from the 60–64 age group to the 70–74 age group in women and from the 60–64 age group to the 75–79 age group in men. This suggests that functional decline becomes more pronounced during the 70s decade in older adults.

When comparing the performance of the subjects in our sample on the Up&Go test with the reference values provided by Rikli and Jones [8], both men and women in our sample demonstrate a similar percentile value relative to their age and sex group, approximately around the 50th percentile.

Some researchers have analyzed the relationship between the Up&Go time and the presence of sarcopenia. The predictive value of the Up&Go time for sarcopenia has also been studied. For example, a time longer than or equal to a cutoff of 10.85 s on the Up&Go test predicted sarcopenia with a sensitivity of 67% and a specificity of 88.7% [59]. None of the men in our sample had a value equal to or higher than this cutoff, and 1.1% of the women did. Of these, all were 76 years of age or older.

In a population of 39,519 subjects aged 66 years or older, with a follow up of 5.7 years, subjects who completed the Up&Go test in more than 10 s had a hazard ratio of developing functional dependence of 1.70 (95% CI, 1.45–2.01) [54]. None of the men in our study had a value equal to or greater than 10 s, whereas 2.3% of the women did, of whom 100% were 76 years of age or older and 50% were over 80 years of age. As such, the Mexican women that we studied probably have a higher risk of developing functional dependence.

In this test, the distance of displacement is constant, so employing a longer time to complete it indicates a poorer performance. When comparing the evolution of performance in this test over the years from the age of 60 onward, women in our sample show a higher annual percentage increase of 2.85%/year, compared to the increases reported in other studies: 2.19% [44], 1.24% [47], 2.15% [15], 1.38% [13], or 1.7% [45], only lower than the study that reports a 6.71% increase in their sample [21]. Men in our sample show a 1.83%/year increase, compared to the increases reported in other studies: 1.33% [44], 1.54% [47], 1.48% [13], or 1.47% [45], only lower than the ones reporting a 2.94% increase [15] or a 5.02% increase [21] in their respective samples. The 2-minute step test (2St) is a test introduced by Rikli and Jones in the SFT battery to assess the aerobic component of physical fitness [60]. However, it does not measure a single specific physical quality but rather evaluates the overall functional capacity in which several components of physical fitness are involved, including the strength of the lower limbs, cardiopulmonary endurance, and the flexibility of the hips and knees. In fact, it has been extensively used in the functional assessment of non-older populations, including individuals with conditions such as osteoarthritis [61], lower-back pain, spinal-cord injury [62], or multiple sclerosis [63], and it can be useful for monitoring the evolution of functional capacity during a general rehabilitation process. However, this test is frequently used in the evaluation of functional capacity in older individuals, both with and without disabilities. Although some age groups of women exhibit significantly lower performance compared to men, there are no significant differences between the sexes when considering the percentiles of individuals from different age and sex groups based on the reference values provided by Rikli and Jones [8]. The percentile performance in this test for all age and sex groups is around the 70th percentile, indicating that the performance is above the mean of their respective reference groups. According to Rikli and Jones [8], this may mean that it has a good relative functional capacity.

The number of step cycles decreases with aging, and taking the value shown by the subjects at 60 years of age as a reference, women and men experienced annual losses of 0.82% and 0.67%, respectively. Women exhibited a similar annual percentage loss, as reported in some studies (−0.82, [64]; −0.85, [45]), or lower than another study showing a loss of −1.10% [44]. Men in the sample had a similar loss, as reported in a publication of −0.63% [45], and lower than another study presenting a loss of −1.25% [15].

Flexibility, in general, is considered one of the important aspects of physical fitness related to health. However, in recent years, whether flexibility can be considered as important as muscular strength, muscular endurance, or cardiopulmonary fitness has been called into question because, among other reasons, unlike the other related factors, no evidence exists that it has a significant predictive value of mortality [65]. Flexibility has uncertain predictive value for the appearance of back pain or injuries in adults. In contrast to the objections or limitations of its determination, flexibility forms part of the classic batteries for the evaluation of the physical condition of older adults [8]. The women in this study, up to the age of 80 years, performed statistically significantly better than the men in the study in the chair sit-and-reach test (lumbar mobility) and in the back-scratch test (shoulder mobility). These sex differences have repeatedly arisen in studies comparing older men and women [15,20] and are often present in younger populations [66]. However, when obtaining the percentile values of each subject in our study based on the references provided by Rikli and Jones [8], we can observe that there are no differences between sexes. Both men and women in our study show scores on the CSRt test around the reference mean in all age groups. However, in the BSt test, our population demonstrates a low level of flexibility.

In men, we found no statistical significance between age and the CSRt result, but in women, we noted a loss of 3.89% per year. In the BSt test, the annual percentage loss was 7.05% in women and 3.51% in men. The decline in performance in tests assessing flexibility is challenging to interpret, particularly in the older adult population. This decline is influenced not only by reduced tissue flexibility due to fibrosis but also by shoulder pathologies that significantly affect the results. Additionally, conditions such as hip or knee osteoarthritis [67,68,69] show a significant increase in prevalence with age. It is noteworthy that studies have reported a 40% prevalence of complete rotator cuff tears [70] and knee osteoarthritis in the population aged over 75 years.

One of the objectives of this study was to obtain reference values for some of the tests most commonly used in the assessment of physical fitness in the aged population. Table 7 shows the cutoff points by age group and sex, which serve to establish cutoffs for older people for each of the tests at the quartile level.

This is the first study to provide reference values for the main SFT battery tests for a population of functionally independent older adults, both men and women, who are noninstitutionalized in Mexico. The reference values obtained in our study can be valuable for caregivers of the elderly, enabling them to evaluate the functional capacity of their patients in relation to their peers and guide the development of tailored care plans and interventions to optimize the well-being of older adults and detect the risk of functional dependence. Our study can serve as a catalyst for future research in other states of Mexico, allowing for data collection from multiple regions of the country and the establishment of more generalizable reference values for functional capacity for the entire older adult population in Mexico, similar to those in many other countries.

However, our study also presents limitations that need to be considered when interpreting and extrapolating the reference values we provide. Firstly, like all cross-sectional studies that offer reference values, it is assumed that the current older adult population will experience the same evolutionary process as the older adults included in the study, which may limit the generalization of the findings to future cohorts of older adults. Secondly, the study sample was from a single region in Mexico (northwest), consisting of noninstitutionalized individuals attending municipal social clubs for older adults, all from an urban area, with no representation of older adults from rural areas. Thirdly, there was a significant disparity in the number of individuals in each age group, and this can introduce biases and limit the ability to draw precise conclusions about specific age cohorts. Fourthly, comparing data values between studies conducted by different researchers in different countries, with variations in methodology and with ethnic and cultural differences, should be approached with caution.

Further research with larger and more diverse samples, including rural and institutionalized older adults, would be valuable in addressing these limitations and providing a more comprehensive understanding of reference values for the older adult population in Mexico.

## 5. Conclusions

This is the first study conducted in a region of Mexico that provides reference values for physical capacity measured by the Senior Fitness Test Battery. Women and men of a similar age show a similar functional level in relation to the reference values provided by the SFT Battery. Women over 80 years of age may be at greater risk of developing functional dependence than men of the same age. Although functional decline is not uniform across different components of physical fitness, it generally approximates a loss of 1% per year from the age of 60.

## Figures and Tables

**Figure 1 healthcare-11-01733-f001:**
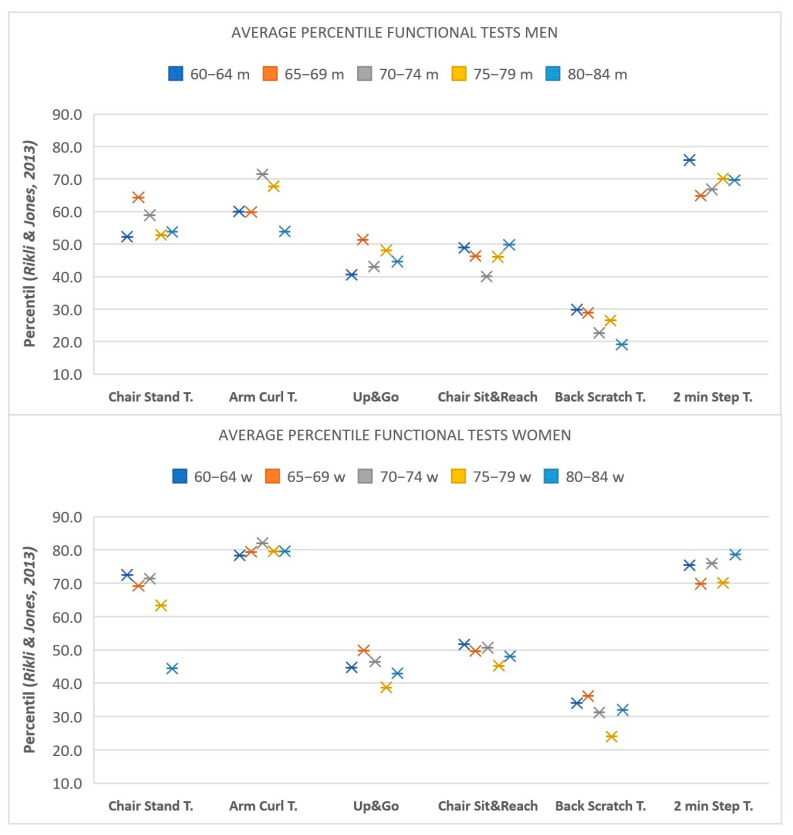
Average percentile for each age group of men and women according to the reference values of Rikli and Jones (2013) [8].

**Table 1 healthcare-11-01733-t001:** Anthropometric variables by sex and age group.

Age Groups	60–64 (a)	65–69 (b)	70–74 (c)	75–79 (d)	80–84 (e)
Women					
n (%)	101 (26)	107 (28)	92 (24)	52 (14)	32 (8)
Height (m)	1.56 ± 0.6 ^d,e,^*	1.55 ± 0.6 *	1.54 ± 0.05 *	1.53 ± 0.06 *	1.52 ± 0.05 *
Weight (kg)	72.4 ± 13.4 ^e,^*	70.1 ± 15.0 ^e,^*	68.6 ± 11.2 *	66.2 ± 12 *	60.9 ± 14.7
BMI	29.1 ± 4.8 ^e^	28.6 ± 5.0	29.3 ± 3.9 *	28.1 ± 4.8	26.7 ± 4.9 *
% Fat	39.4 ± 6.6 ^e,^*	39.9 ± 6.5 ^e,^*	39.2 ± 5.4 ^e,^*	36.8 ± 6.7 *	34.4 ± 9.3 *
Men					
n (%)	39 (24)	56 (34)	29 (18)	29 (18)	12 (8)
Height (m)	1.68 ± 0.06	1.7 ± 0.1	1.68 ± 0.06	1.67 ± 0.06	1.65 ± 0.06
Weight (kg)	81 ± 17.7 ^e^	77.8 ± 13.8	77.1 ± 14.3	73.6 ± 14.2	65.5 ± 12.6
BMI	27.6 ± 7.6 ^e^	26.4 ± 6.5 ^e^	24.7 ± 5.9	25.4 ± 7.9	24.9 ± 7.7
% Fat	28.1 ± 8.2	26.4 ± 7.8	25.6 ± 7.2	27.1 ± 11.5	24.9 ± 7.7

Mean ± standard deviation; * = *p* < 0.05 with respect to the same age group of the other sex. Each age group has been assigned a letter in parentheses, and the superscripts in the table indicate significant differences between the group in the column and the groups of the same sex indicated by the corresponding letters in the post hoc ANOVA analysis.

**Table 2 healthcare-11-01733-t002:** Men and women with low, normal, overweight, or obese body weight.

	Low		Normal		Overweight		Obesity	
Age Group	%	n	%	n	%	n	%	n
Women								
60–64	0	0	16.8	17	41.6	42	41.6	42
65–69	1.9	2	21.5	23	38.3	41	38.3	41
70–74	3.3	3	15.2	14	37	34	44.6	41
75–79	1.9	1	26.4	14	37.7	20	34	18
80–84	9.4	3	28.1	9	37.5	12	25	8
All	2.3	9	20	77	38.7	149	39	150
Men								
60–64	0	0	20.5	8	41	16	38.5	15
65–69	5.4	3	28.6	16	41.1	23	25	14
70–74	3.4	1	34.5	10	34.5	10	27.6	8
75–79	6.9	2	31	9	44.8	13	17.2	5
80–84	8.3	1	66.7	8	16.7	2	8.3	1
All	4.2	7	30.9	51	38.8	64	26.1	43

% = proportion of the age subsample of each sex; n = number of cases.

**Table 3 healthcare-11-01733-t003:** Evaluation of strength, sex, and age group.

Age Groups	60–64 (a)		65–69 (b)		70–74 (c)		75–79 (d)		80–84 (e)	
	X ± DS	*Pr*	X ± DS	*Pr*	X ± DS	*Pr*	X ± DS	*Pr*	X ± DS	*Pr*
Women										
n (%)	101 (26)		107 (28)		92 (24)		52 (14)		32 (8)	
Handgrip T.	23.5 ± 4.8 ^c,d,e,^*		22.5 ± 3.5 ^d,e,^*		21.3 ± 3.9 ^d,e,^*		19.7 ± 4.3 *		18.1 ± 3.9 *	
Arm Curl T.	20.5 ± 3.7 ^d,e^	*77.8 ± 19.8* *	19.5 ± 3.1 ^e^	*79.9 ± 17.4* *	19.2 ± 3.1 ^e^	*82.1 ± 15.4* *	18.3 ± 2.9	*79.6 ± 14.5* *	17.3 ± 3.5	*79.6 ± 19.2* *
Chair Stand T.	17.9 ± 4.0 ^b,c,d,e^	*72.1 ± 25* ^e,^*	15.8 ± 3.7 ^d,e,^*	*68.5 ± 27.8* ^e^	15.5 ± 3.2 ^e^	*71.4 ± 22.3* ^e,^*	14.1 ± 3.0 ^e^	*63.3 ± 24.0* ^e^	10.2 ± 3.4	*44.4 ± 25.5*
Men										
n (%)	39 (24)		56 (34)		29 (18)		29 (18)		12 (8)	
Handgrip T.	37.7 ± 7.7 ^c,d,e^		36.5 ± 6.1 ^d,e^		34.1 ± 6.5		31.2 ± 6.8		26.9 ± 4.8	
Arm Curl T.	20.7 ± 3.4 ^e^	*62.2 ± 22.3*	19.8 ± 3.8 ^e^	*59.9 ± 24.1*	20.1 ± 2.9 ^e^	*68.1 ± 17.8*	19.1 ± 4.3	*67.8 ± 24.5*	16.5 ± 3.3	*53.9 ± 24.5*
Chair Stand T.	16.5 ± 3.6 ^e^	*52.3 ± 26.1*	17.3 ± 3.5 ^d,e^	*64.3 ± 22.1*	15.6 ± 3.3	*58.9 ± 24.7*	14.4 ± 4.6	*52.8 ± 32.1*	12.7 ± 3.8	*53.8 ± 30.4*

Mean ± standard deviation; * = *p* < 0.05 with respect to the age group of the other sex. Each age group has been assigned a letter in parentheses, and the superscripts in the table indicate significant differences between the group in the column and the groups of the same sex indicated by the corresponding letters in the post hoc ANOVA analysis. Pr = (indicated with italic letter), mean percentile based on reference values published by Rikli and Jones 2013) [8]. Units used are kilograms for handgrip, and number of repetitions for arm curl and chair stand.

**Table 4 healthcare-11-01733-t004:** Evaluation of UGt by sex and age group.

Age Groups	60–64 (a)		65–69 (b)		70–74 (c)		75–79 (d)		80–84 (e)	
	X ± DS	*Pr*	X ± DS	*Pr*	X ± DS	*Pr*	X ± DS	*Pr*	X ± DS	*Pr*
Women										
n (%)	101 (26)		107 (28)		92 (24)		52 (14)		32 (8)	
Up&Go	5.4 ± 0.9 ^c,d,e^	*44.7 ± 23.1*	5.7 ± 0.9 ^d,e,^*	*49.4 ± 24.6*	6.1 ± 1.0 ^d,e^	*46.0 ± 23.4*	7.7 ± 2.8	*38.7 ± 26.5*	8.6 ± 3.7	*42.9 ± 26.9*
2 min Step T.	115.6 ± 27.3 ^d,e,^*	*75.2 ± 23.5*	114 ± 32.3	*71 ± 24.6*	106.8 ± 27.8	*73.6 ± 25.3*	98.7 ± 20.7 *	*68.3 ± 25*	93.7 ± 26.4	*73.3 ± 29.2*
Men										
n (%)	39 (24)		56 (34)		29 (18)		29 (18)		12 (8)	
Up&Go	5.2 ± 1.1 ^d,e^	*40.6 ± 22.7*	5.1 ± 0.7 ^d,e^	*51.4 ± 19.6*	5.7 ± 1.1 ^e^	*43.0 ± 25.9*	6.1 ± 1.6	*48.0 ± 26.0*	6.8 ± 1.3	*45.6 ± 24.1*
2 min Step T.	127.5 ± 38.9	*74.5 ± 30.5*	118.1 ± 32.9	*65.6 ± 31.7*	112 ± 32.5	*68 ± 29.5*	118 ± 42.9	*71.4 ± 27.4*	97.1 ± 22.2	*64.3 ± 29.2*

Mean ± standard deviation; * = *p* < 0.05 with respect to the age group of the other sex. Each age group has been assigned a letter in parentheses, and the superscripts in the table indicate significant differences between the group in the column and the groups of the same sex indicated by the corresponding letters in the post hoc ANOVA analysis. Pr = (indicated with italic letter), mean percentile based on reference values published by Rikli and Jones 2013) [8]. Units used are seconds for the Up&Go test, and number of repetitions for the 2minSt.

**Table 5 healthcare-11-01733-t005:** Evaluation of flexibility.

Age Groups	60–64 (a)		65–69 (b)		70–74 (c)		75–79 (d)		80–84 (e)	
	X ± DS	*Pr*	X ± DS	*Pr*	X ± DS	*Pr*	X ± DS	*Pr*	X ± DS	*Pr*
Women										
n (%)	101 (26)		107 (28)		92 (24)		52 (14)		32 (8)	
Chair Sit and Reach	5.9 ± 10.2 ^d,^*	*52.2 ± 29.3*	5.1 ± 7.5 *	*49.6 ± 25.8*	3.0 ± 8.8 *	*49.5 ± 27.8*	0.4 ± 9.4 *	*44.4 ± 28.5*	0.7 ± 8.8	*48 ± 29*
Back-Scratch T.	−8.1 ± 10.5 ^d,e,^*	*33.7 ± 30.8*	−8.8 ± 11.8 ^d,^*	*36.5 ± 31.0*	−10.9 ± 10.5 *	*32 ± 30.1*	−15.8 ± 10.7 *	*24 ± 26.8*	−14.5 ± 12.0 *	*32 ± 27.7*
Men										
n (%)	39(24)		56(34)		29(18)		29(18)		12(8)	
Chair Sit and Reach	0.5 ± 12.3	*48.9 ± 30.3*	−2.1 ± 10.1	*45.4 ± 27*	−4.2 ± 10.7	*40.1 ± 28.3*	−4.6 ± 10.7	*46.1 ± 27.8*	−5.8 ± 13.8	*49.8 ± 32*
Back-Scratch T.	−18.2 ± 13.4	*29.8 ± 29.4*	−19.9 ± 11.2	*28.3 ± 23.6*	−24.6 ± 11.6	*21.8 ± 23.2*	−24.7 ± 10.7	*26.5 ± 22.4*	−28.9 ± 10.4	*19.1 ± 18.1*

Mean ± standard deviation; * = *p* < 0.05 with respect to the age group of the other sex. Each age group has been assigned a letter in parentheses, and the superscripts in the table indicate significant differences between the group in the column and the groups of the same sex indicated by the corresponding letters in the post hoc ANOVA analysis. Pr = (indicated with italic letter), mean percentile based on reference values published by Rikli and Jones 2013) [8]. Unit used is centimeters.

**Table 6 healthcare-11-01733-t006:** Percentage loss per year during adulthood starting at 60 years of age.

		All				Women				Men			
		a	b	r^2^	SEE	a	b	r^2^	SEE	a	b	r^2^	SEE
Annual % decrease	Handgrip T.	−1.18	70.52	0.69	5.89	−1.14	68.82	0.72	5.39	−1.36	84.42	0.78	5.39
	Arm Curl T.	−0.74	44.77	0.70	3.68	−0.68	41.38	0.55	4.69	−0.76	44.79	0.36	7.62
	Chair Stand T.	−1.78	113.84	0.78	7.02	−1.93	121.43	0.73	8.74	−1.32	89.03	0.52	9.50
	Up&Go	2.42	−152.04	0.81	8.86	2.85	−180.09	0.72	13.54	1.83	−115.10	0.63	10.44
	2 min Step T.	−0.85	56.91	0.57	5.57	−0.82	56.03	0.29	9.68	−0.67	41.18	0.28	8.14
	Chair Sit and Reach	−0.42	10.61	0.07	11.91	−3.89	212.66	0.44	33.13	−6.40	286.80	0.06	182.69
	Back-Scratch T.	5.26	−322.22	0.55	35.82	−7.05	−445.69	0.50	52.69	−3.51	−196.98	0.20	52.25

a = slope; b = intercept; r^2^ = regression coefficient; SEE = standard error of estimation.

**Table 7 healthcare-11-01733-t007:** Reference values of the components of physical fitness.

Grup	Centil	Handgrip T.	Arm Crul T.	Chair Satnd T.	Up&Go	2 min Step T	Chair Sit-Reach	Back Scratch T.
Women 60–64	10th	17	15	13	4.4	88	−9	−21
	25th	20	18	15	4.7	98	0	−16
	50th	23	21	18	5.3	112	6	−7
	75th	26	23	21	5.9	129	13	2
	90th	29	26	24	6.8	149	17	4
Man 60–64	10th	27	16	11	4.3	86	−20	−35
	25th	34	18	14	4.5	105	−9	−28
	50th	39	20	16	4.9	122	1	−21
	75th	43	23	19	5.6	148	11	−10
	90th	49	24	21	6.7	158	15	3
Women 65–69	10th	18	15	11	4.6	79	−6	−26
	25th	20	17	13	5.0	92	0	−16
	50th	22	19	16	5.5	108	5	−9
	75th	25	22	18	6.3	124	11	0
	90th	27	24	21	6.8	150	16	5
Man 65–69	10th	27	14	13	4.0	77	−16	−31
	25th	33	17	14	4.6	93	−11	−27
	50th	36	20	17	5.0	115	0	−18
	75th	40	23	19	5.6	135	5	−14
	90th	44	25	23	5.9	160	10	−2
Women 70–74	10th	17	15	11	5.0	81	−9	−24
	25th	19	17	13	5.4	93	0	−18
	50th	22	19	16	5.9	106	4	−12
	75th	24	22	17	6.7	120	9	−3
	90th	26	23	20	7.5	136	15	4
Man 70–74	10th	25	18	12	4.4	79	−19	−35
	25th	30	18	13	4.8	95	−13	−30
	50th	33	20	16	5.5	102	−2	−27
	75th	37	22	19	6.4	132	3	−17
	90th	40	24	20	7.6	160	9	−7
Women 75–79	10th	15	15	10	5.2	76	−13	−29
	25th	17	16	12	5.9	86	−8	−25
	50th	19	18	14	7.0	99	3	−17
	75th	21	20	16	7.8	111	8	−8
	90th	24	22	19	9.6	122	12	1
Man 75–79	10th	23	13	9	4.6	74	−23	−40
	25th	26	16	10	4.8	94	−11	−34
	50th	32	19	13	5.9	112	−4	−24
	75th	34	21	19	7.3	126	3	−17
	90th	41	25	20	8.9	150	12	−12
Women 80–84	10th	13	13	5	5.5	71	−12	−34
	25th	15	14	8	6.2	87	−7	−22
	50th	18	18	11	7.9	100	0	−14
	75th	21	20	12	9.0	111	8	−7
	90th	22	22	14	10.8	126	13	1
Man 80–84	10th	18	12	7	5.1	70	−27	−48
	25th	25	14	9	5.9	85	−19	−32
	50th	27	16	13	6.3	106	−3	−28
	75th	31	19	16	8.2	115	5	−25
	90th	32	22	18	8.9	120	14	−11

Units used are kilograms for the handgrip strength test; number of repetitions for the arm curl, chair stand, and 2 min step tests; seconds for the Up&Go test; and centimeters for the chair sit–reach and back-scratch tests.

## Data Availability

Additional information is available upon request from the senior author, japazf@unileon.es.

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
