# Peer review of "Functional Capacity of Noninstitutionalized Older Adults from Northwest Mexico: Reference Values"

_healthcare, 2023, doi:10.3390/healthcare11121733_

Round 1

Reviewer 1 Report

pleae refer to the folowing reference. seems duplicate. 

Association between physical activity and physical and functional performance in non-institutionalized Mexican older adults: a cohort study

Brenda María Martínez-Hernández,1 Oscar Rosas-Carrasco,2 Miriam López-Teros,2 Alejandra González-Rocha,3 Paloma Muñoz-Aguirre,4 Rosa Palazuelos-González,5 Araceli Ortíz-Rodríguez,6 Armando Luna-López,7 and Edgar Denova-Gutiérrezcorresponding author3

Author Response

REVIEWER 1:

Authors’ Response (AR): Dear reviewer, we found your comments positive, and we thank you for the time you have spent reviewing our article, and we express our gratitude, thank you very much.

Reviewer's comments:

 1.Please refer to the folowing reference. seems duplicate.

  1. (AR): Thank you very much for your comment, we appreciate your highlighting that article in case we had missed it.

But the article you are referring to is an article carried out by other researchers, in another city (Mexico City), with a different population (300 people, with a minimum selection age of 50 years), a longitudinal study with objectives clearly different from ours, with different study variables and different measurement instruments.. The cited study and ours have only one element in common, and that is that of all the variables analyzed, only one has been evaluated in both studies: the sitting and getting up from the chair test.  Unlike the study you mention, we have not used geriatric assessment scales (Barthel Index, Lawton and Brody scale measures the instrumental activities of daily living (IADL)), nor did we assess physical condition or its components with the Short Physical Performance Battery (SPPB), which only has in common with the Senior Fittnes Test the test of sitting down and getting up from a chair, whose values are not shown in the aforementioned study because they form part of the total score of the SPPB.

The aforementioned study does not disaggregate the sit-to-stand test by age group, nor does it establish reference values.

Nor is the statistical treatment the same in both studies.

Therefore, we sincerely appreciate your observation, but we must point out that the studies are not similar, nor are they duplicated or replicated, and in fact the article you mention is not mentioned in our study because it does not serve to contrast our data.

Reviewer 2 Report

Appropriate subject matter, and with a simple introduction that states the problem to be addressed through the stated objective.

However, there are many aspects to take into account that make me have my doubts about whether the article should be publishable. 

ABSTRACT

- I would include the date of patient recruitment and assessment in the abstract.

- I would not use terms such as better to refer to strength values, but for example greater or lesser but not better or worse.

METHODS

- Were patients with medication for pathologies excluded? 

- Normally the handgrip strenght is made for an average of 3 attempts, why was it decided to make 4 attempts?

- For the handgrip there is a formula that determines, according to the size of the hand, at what distance the device should be extended for the patient's grip. has this been done differently? has it been forgotten? is there another different protocol? It should be justified and indicated with bibliographic reference.

- Why did you decide to divide the patients into these age ranges and not others?

RESULTS

- First paragraph: Whenever it is mentioned whether or not there are differences between groups, it should be specified whether it is significant or non-significant in the text, without the need to go to the table.

- Apply to the rest of the results paragraphs.

- Table 1 lacks the n for the number of patients in each group.

- Table 1 does not specify the P value in the table footer, i.e. the significance level.

- In some tables men/women is in lower case and in others in upper case, unify in all tables (example: table 3 and table 4).

DISCUSSION

- The discussion should reflect, in different paragraphs of the discussion and as a closure of ideas, what clinical implication has for other researchers, future studies or health personnel, the fact that what is being discussed occurs or the data obtained.

- The fact that in the discussion mention is made of tables describing results makes the discussion lose its own sense, since this should be done in the results, the discussion section being a section where the results obtained are contrasted with those found by other authors and the possible reasons for the differences or coincidences.

- I suggest rewriting the discussion where ideas are better contrasted (in some cases the discussion becomes difficult to read and tires the reader). Summarize and be concise.

- Figures should be part of the results

- Only one limitation is mentioned and the study has more, list them.

- List more strengths of your study (only 1 is mentioned).

BIBLIOGRAPHY

- There is one reference from 1985 and several from 1999, why? is there justification? could more current ones be used?

Must be revised.

Author Response

Authors’ Response (AR): Dear Reviewer, thank you for giving us the opportunity to submit a revised draft of the manuscript. We appreciate the time and effort that you and other reviewers have dedicated to providing your valuable feedback on our manuscript. We are grateful to the reviewers for their insightful comments on the manuscript. We have responded point by point to all of your accurate corrections and pertinent observations. In the reply we do not indicate the line number, as the document has been extensively rewritten. We hope that you will find that we have taken advantage of your indications, which again we sincerely thank you for. We believe they have helped us to substantially improve the article.

Reviewer's comments:

ABSTRACT

1. I would include the date of patient recruitment and assessment in the abstract.

1. (AR): Line (xx): Following your recommendation, we have added:

Was carried out over a period of 6 months, from january 2019 to june 2019.  (abstract and methodology). We have modified the entire abstract, and have rewritten it again.

2. I would not use terms such as better to refer to strength values, but for example greater or lesser but not better or worse.

2. (AR): We have replaced in the text the word better by greater.

METHODS

  1. Were patients with medication for pathologies excluded?

3.- (AR): The large majority of older adults take one or more medications. Patients were not excluded on the basis of the medication they were taking, but rather on the presence of chronic or acute uncontrolled diseases. It would be difficult to obtain a large sample of older adults if any medication was used as an exclusion criterion.

4. Normally the handgrip strenght is made for an average of 3 attempts, why was it decided to make 4 attempts?

4.- (AR): We appreciate your sharpness of comment. In fact three repetitions are often performed to obtain the manual grip strength. We evaluate both hands, to avoid confusion between dominant hand and strong hand, because it is frequent the presence of some degree of injury, for example by the presence of osteoarthritis in the hands in older people with different degree of involvement between the hands.

In addition, to avoid fatigue and to make the evaluation more fluid, the evaluation was performed on each hand twice and the highest value of the four measurements was used. We agree with the reviewer's assessment, but we would also like to point out that in the meta-analysis, which we indicate in the following point, it can be seen that studies in which the pressure force is assessed twice are not uncommon either.  Taking into account the size of the sample and that in our study the grip strength variable is a variable for establishing baseline values in a large sample, and is not an outcome variable after an intervention, we do not believe that this detail does not affect the validity of our study.

 5. For the handgrip there is a formula that determines, according to the size of the hand, at what distance the device should be extended for the patient's grip. has this been done differently? has it been forgotten? is there another different protocol? It should be justified and indicated with bibliographic reference.

5.- (AR): Thank you for the observation that we will keep in mind in the future. We were unaware that a formula existed and was applied to establish the length between the hand dynamometer grip handles in elderly people, but we will certainly keep it in mind for future studies. The procedure we followed consisted of adjusting the distance of the dynamometer handles until the subject felt comfortable with the handgrip to perform the gripping force (within the range of handle variation allowed by the hydraulic hand dynamometer). (Jamar J00105).

In one of the systematic review articles we had consulted and had not placed in the bibliography, "Measurement of hand grip strength in the elderly: A scoping review with recommendations", doi:10.1016/j.jbmt.2019 .05.029 , under "Dynamometer handle setting", it states: "This review has found that dynamometer handle setting is often not taken into account when assessing hand grip in older adults and therefore we recommend that the individual choose the grip position that best suits their hand and comfort."

In response to your comment and so as not to confuse future readers, we have added this point in the methodology, and also added the bibliographic citation, which reads as follows  "with a position of the dynamometer handles with which the individual feels comfortable in the grip" (DOI: 10.1016/j.jbmt.2019.05.029).

  1. Why did you decide to divide the patients into these age ranges and not others?

6.- (AR): We performed previous analyses establishing the groups by years of age, by groups of 2 years, etc.  In general, for each of the variables there is a large dispersion (standard deviation) in people of the same age, and also in groups aggregated by two years. To simplify the presentation of results and to smooth the dispersion (standard deviation), we use groups of 5 years, which is the form most frequently used in studies that provide normative values of strength in older people. (vg: DOI: 10.2519/jospt.2018.7851; PMID: 27759870; DOI: 10.1177/1753193408096763; DOI: 10.11622/smedj.2015164).

RESULTS

  1. First paragraph: Whenever it is mentioned whether or not there are differences between groups, it should be specified whether it is significant or non-significant in the text, without the need to go to the table.

7.- (AR): Thanks for the comment, we have added the adjective "significant" to the words "difference" or "differences" throughout the text.

  1. Apply to the rest of the results paragraphs.

8.- (AR): Thank you for your comment, we have also added it to the results section.

  1. Table 1 lacks the n for the number of patients in each group.

9.- (AR): Thank you for your comment, we have added it to the tables.

  1. Table 1 does not specify the P value in the table footer, i.e. the significance level.

10. (AR): We have replaced "* = Statistical significance" with "* = p<0.05 Statistical significance” at the bottom of the tables.

11. In some tables men/women is in lower case and in others in upper case, unify in all tables (example: table 3 and table 4).

11. (AR): Thank you for the correction, we have unified it in all the tables.

 DISCUSSION

  1. The discussion should reflect, in different paragraphs of the discussion and as a closure of ideas, what clinical implication has for other researchers, future studies or health personnel, the fact that what is being discussed occurs or the data obtained.
  2. The fact that in the discussion mention is made of tables describing results makes the discussion lose its own sense, since this should be done in the results, the discussion section being a section where the results obtained are contrasted with those found by other authors and the possible reasons for the differences or coincidences.
  3. I suggest rewriting the discussion where ideas are better contrasted (in some cases the discussion becomes difficult to read and tires the reader). Summarize and be concise.

12,13,14 (AR): We also believe that these three observations you have made are very appropriate and pertinent to make the article easier to read, more orderly, and more appropriate to the discussion format. We have rewritten the entire discussion chapter. Thank you again for your help in improving the article.

  1. Figures should be part of the results:

15 (AR): We have moved figure 2 to the results section (now it is figure 1), figure 1 of the original article has been removed (as well as the explanation or discussion of it).

  1. Only one limitation is mentioned and the study has more, list them.
  2. List more strengths of your study (only 1 is mentioned).

16, 17 (AR): We have modified by adding more strengths and more limitations of the study.

BIBLIOGRAPHY

18.There is one reference from 1985 and several from 1999 IMC, why? is there justification? could more current ones be used?

18 (AR): We have reviewed the bibliography and we think that 1985 is relevant, since it is the first author who collects in a publication a previous consensus made for the concept of physical capacity. And the publication of the Sernior Fitness Test battery, which is the one we have used, was in 1999. We believe that we have made an adequate bibliographic review and we have tried to reference the most relevant articles.

Reviewer 3 Report

First of all, congratulations for the work done, then I will mention a number of changes and recommendations in order to obtain clearer and more accurate information.

- Comments on the introduction:

·       The introduction is very clear and well conducted, congratulations, I only want to make one remark.

·       I think that the objective, should be rewritten as this one: “The aim of this study was to describe the evolution of the main components of physical capacity throughout the aging process in older men and women in different age groups to provide reference values of the functional capacity for the older adult population of northwestern Mexico”; I think that there is only one objective.

- Comments on material and methods

·       “The normality of the distribution was verified with the Kolmogorov–125 Smirnov test”, this test assesses the normality of the distribution of the residuals, you should point out this one.

·       The same when you talk about normal distribution and non-normal distribution in lines 133 and 134.

- Comments on results:

  • You should revise table 1 format, WOMEN is written in bold type and MEN not, please use the same style in all table and titles, there is a line under HEIGHT that should be erased.
  • Line 163, you should write “nor in men”.
  • Table 3, choose between WOMEN and MEN, or Women and Men, and follow the same style in all tables.
  • Check all tables format, and use the same in all of them.

- Comments on discussion:

-        Figure 2, you use “WOMEN” and “MEN” in all the manuscript, why did you use males in this figure?

-        Figure 2 footer, there is a point that is not needed between according and to: “Average percentile for each age group of men and women according.to Rikli and Jones 359 reference values, (2013).”

Author Response

Authors' Response (AR): Dear Reviewer, Thank you for giving us the opportunity to submit a revised draft of the manuscript. We appreciate the time and effort you and have spent in providing your valuable comments on our manuscript. We are grateful for the help you have given us with your pertinent comments and corrections. We hope that we have sufficiently addressed your comments. In the reply we do not indicate the line number, as the document has been extensively rewritten.

Reviewer's comments:

INTRODUCTION

  1. I think that the objective, should be rewritten as this one: “The aim of this study was to describe the evolution of the main components of physical capacity throughout the aging process in older men and women in different age groups to provide reference values of the functional capacity for the older adult population of northwestern Mexico”; I think that there is only one objective.

1.- (AR): We have carefully considered the reviewer's suggestion, but respectfully, we believe it is appropriate to maintain the objectives, as we do analyze the loss of functions throughout age (Table 6) and also provide reference values, now centiles in Table 7.

MATERIAL AND METHODS

  1. “The normality of the distribution was verified with the Kolmogorov– Smirnov test”, this test assesses the normality of the distribution of the residuals, you should point out this one.

2.-(AR): Thank you for your keen observation. Indeed, the study of the normality of the distribution of quantitative variables was performed using the Kolmogorov-Smirnov test, by calculating distribution parameters from the sample data, employing both the asymptotic results and the Lilliefors correction. To make it suitable for expert readers, we have modified the text to leave it as follows: “The normality of the distribution was verified with the Kolmogorov–Smirnov test with a Lilliefors significance correction”

  1. The same when you talk about normal distribution and non-normal distribution in lines 133 and 134.

3. (AR): Having clarified the procedure used to determine normality, we believe that in this sentence it can be maintained without leading to confusion.

RESULTS

  1. You should revise table 1 format, WOMEN is written in bold type and MEN not, please use the same style in all table and titles, there is a line under HEIGHT that should be erased.

4. (AR): Thank you very much for your correction, it has been corrected.

5. Line 163, you should write “nor in men”.

5. (AR): Thank you very much for your correction, it has been corrected.

6. Table 3, choose between WOMEN and MEN, or Women and Men, and follow the same style in all tables.

6.-(AR): Thank you very much for your correction, it has been corrected.

  1. Check all tables format, and use the same in all of them.

7.- (AR): Thank you very much for your correction, it has been corrected.

DISCUSSION:

  1. Figure 2, you use “WOMEN” and “MEN” in all the manuscript, why did you use males in this figure?

8.- (AR): Thank you very much for your correction, it has been corrected, (now is figure 1). The figure 1 of the original article has been removed (as well as the explanation or discussion of it).

  1. Figure 2 footer, there is a point that is not needed between according and to: “Average percentile for each age group of men and women according.to Rikli and Jones 359 reference values, (2013).”

9.- (AR): Thank you very much for your correction, it has been corrected

Reviewer 4 Report

I would like to congratulate the authors for the work done and for bringing this topic to the special issue. Unfortunately, the article has a significant series of flaws that make it unpublishable in its current state. I will provide a series of recommendations to improve the article so that it can be publishable in the future.

ABSTRACT

The abstract talks little about the study's results, making it unclear from reading what has been found. Please rephrase this part to make it more explanatory (take an example for reference 23)

INTRODUCTION

32-40: I understand that the article is about the Mexican population, but it would be good to contextualize the data provided in a global or, at least, regional setting (for example, Latin America, Central America...) depending on data availability. And then talk about Mexico per se.

41-46: I don't understand how this section fits into the article. Consider removing it or, if not, justify it and rephrase it.

47-53: Again, I don't understand how this part is written. I would also integrate how functional capacity is an important component of intrinsic capacity and the WHO's call to assess it through ICOPE.

MATERIAL AND METHODS

Starting from table 3, the results are compared with Rikli and Jones [8], which has not been explained in the methodology of the study or justified in the introduction or any other section of the text. Please explain and modify the article in this way.

RESULTS

206-213: here, you are talking about methods, please change it

As you mentioned in the introduction: "The aims of this study were to describe the evolution of the main components of physical capacity throughout the aging process in older men and women in different age groups to provide reference values of the functional capacity for the older adult population of northwestern Mexico". However, while the evolution can be seen, the normative values are not provided, only table 7.

DISCUSSION

The discussion is written in a disjointed manner, with ideas jumping from one side to the other. I think it would be convenient to completely restructure this section.

223-228: The discussion section cannot start by recounting the study's objectives. This has already been stated. It should highlight the main findings of the results and there is no mention of the normative valuales (the other objective of the study).

229-235: What you are explaining here is a limitation of the study, as there is a problem of external validity to the general population. I recommend relocating and rephrase this to a limitations section.

236-251: I don't understand if the aim of the study was functional capacity and normative values, why the discussion is about obesity. Also the meaning of figure 1

256-347: The section mainly discusses aspects that were intended to be evaluated in the study. However, what is done mainly is to comment on them as in the results section without being a real discussion about why these results occur, the previous studies that support or contradict them, and the implications that the results have for the population and/or science.

348-357: Here, table 7 and figure 2 are introduced for the first time, when they should have been introduced in the results section.

361-371: Translation: This is the section on strengths and weaknesses. I have already commented on other weaknesses throughout the review that should be added to this section. It would also be convenient to comment on the sample size (can normative values for the entire northwest of Mexico be established with 550 participants?).

With all the changes mentioned, the conclusions would have to be rewritten

TABLES:

In general, the foot of the tables is confused with the general text of the article. Please modify.

Statistically significant differences are indicated in the tables with an asterisk. However, it is not commented anywhere which p-values will be considered significant or not. Additionally, a p-value of 0.049 is not the same as p<0.00001. Please indicate the p-values of the differences or at least put ranges.

In general, the tables are not referenced in the text. Please correct this.

Table 7. The normative values expressed in 3 quartiles (q1, q2, q3) are not the best way to express these values. Usually, studies of this nature express them in 5 quartiles (for example, reference 23 with 10th, 25th, 50th, 75th, and 90th). Please justify the calculation methodologically in the METHODS section and consider expanding the quartiles for calculation (which I believe is vital to establish normative values).

FIGURES

Figures 1 and 2 appear in the discussion without being mentioned in either the methodology or results section. Please modify this.

Figure 1 missed label of X side

Please avoid the use of elderly

Author Response

Authors’ Response (AR):

Dear reviewer, first and foremost, we would like to express our gratitude for accepting the task of reviewing our manuscript. We sincerely appreciate your effort, time, and the thoroughness of your observations and corrections. Thank you very much. Your feedback, along with that of the other reviewers, has prompted us to make important and extensive changes to our work. We have reworked the abstract, rearranged paragraphs, removed certain comments and a figure, performed new statistical analyses to present reference data in percentiles, reorganized and rewritten the discussion, strengths and weaknesses, and the conclusions. Additionally, your suggestions have compelled us to delve deeper into the literature search for relevant studies from which we could extract comparisons regarding the progression of physical capacity in populations of older adults, specifically those utilizing the Senior Fitness Test battery. We have analyzed the trends in these populations and compared them with our findings. The answer we have given you, we have not indicated the lines in which they are found because the document is largely new. Thanks to your comments, the manuscript has significantly improved, and we have made a concerted effort to address each of your points in a comprehensive manner. We hope you will recognize the dedication and diligence we have demonstrated in addressing your feedback during these past few days.

Reviewer's comments:

ABSTRACT

  1. The abstract talks little about the study's results, making it unclear from reading what has been found. Please rephrase this part to make it more explanatory (take an example for reference 23)

1.- (AR): We have rewritten the abstract, taking into account the suggestions made by the reviewer.

 INTRODUCTION

  1. 32-40: I understand that the article is about the Mexican population, but it would be good to contextualize the data provided in a global or, at least, regional setting (for example, Latin America, Central America...) depending on data availability. And then talk about Mexico per se.

2.- (AR): For better contextualization, we have left the paragraph as follows:

In 2008, Mexico ranked fifteenth in the world in respect of the number of older adults as a percentage of its total population, with a value of 6.1%; by 2020, this pro-portion had reached 12.2%, which shows the accelerated rate of aging of its population.This study was conducted in Hermosillo, the capital of the state of Sonora, in northwestern Mexico, sharing a border to the north with the U.S. state of Arizona and to the west with the Sea of Cortes, being located in the Sonoran Desert; its climate is de-sert-like and extremely hot. According to data from the National Institute of Statistics and Geography (INEGI) in 2020, the estimated population of Hermosillo was approxi-mately 968,000, of which 12% was over 60 years old. In global terms, life expectancy has increased and living conditions have improved in all populations. In Mexico, life expectancy at birth in 2019 was 76 years (78.9 for women and 73.1 for men), and healthy life expectancy was 65.8 years (67.2 years for women and 64.3 for men); these values are lower than the average for the WHO Region of the Americas

3.41-46: I don't understand how this section fits into the article. Consider removing it or, if not, justify it and rephrase it.

3.- (AR): The reason for including it was to emphasize that the concept of older adults is a 'label' that encompasses individuals with a wide range of life stages (from 60 years old until death). Therefore, older adults, for example, at 65 years old, have little in common with those at 90 years old, despite being labeled as 'older adults' in the same way. However, in order to make the article more accessible to readers and in response to their advice, we have removed that paragraph.

  1. 47-53: Again, I don't understand how this part is written. I would also integrate how functional capacity is an important component of intrinsic capacity and the WHO's call to assess it through ICOPE.

4.-(AR): We appreciate your valuable suggestion, and the paragraph has been revised as follows:

“The classic definition of functional physical capacity, as reported by Caspersen et al. (1985), is as follows: “the ability to perform daily tasks with vigor and alertness, without undue fatigue and with ample energy to enjoy leisure-time activities and to cope with unforeseen emergencies”. The main health-related components of functional capacity are strength, aerobic endurance, flexibility, power, speed, agility, balance, and body composition. An adequate functional physical capacity contributes to successful aging, and thus physical capacity assessment is also important in the context of the World Health Organization's Global Strategy and Action Plan on Ageing and Health 2016-2020 as part of comprehensive care for older people “

 MATERIAL AND METHODS

  1. Starting from table 3, the results are compared with Rikli and Jones [8], which has not been explained in the methodology of the study or justified in the introduction or any other section of the text. Please explain and modify the article in this way.

5.-(AR), Thank you very much for your observation. Indeed, we had not included it. We have now added it in the methodology section, resulting in the following: “The percentile was calculated for the value obtained in each of the tests for each participant, by comparing it with the age- and sex-specific normative values obtained from a population of older adults in the United States, as published by the creators of the Senior Fitness Test battery” 

RESULTS

  1. 206-213: here, you are talking about methods, please change it

6.- (AR): We have made changes to the paragraph referring to the "statistical analysis" section and relocated it.

  1. As you mentioned in the introduction: "The aims of this study were to describe the evolution of the main components of physical capacity throughout the aging process in older men and women in different age groups to provide reference values of the functional capacity for the older adult population of northwestern Mexico". However, while the evolution can be seen, the normative values are not provided, only table 7.

7.- (AR): We have conducted and presented the comparison of functional capacity components among age groups, both within the same sex and between sexes, primarily in Tables 3, 4, and 5. These tables present the data separately for men and women, as well as comparisons between age groups. Additionally, Table 6 showcases the longitudinal changes throughout aging, segregated by sex rather than age groups. As suggested, we have included normative values in Table 7, organized in deciles format (10th, 25th, 50th, 75th, and 90th).

 DISCUSSION

  1. The discussion is written in a disjointed manner, with ideas jumping from one side to the other. I think it would be convenient to completely restructure this section.

9.223-228: The discussion section cannot start by recounting the study’s objectives. This has already been stated. It should highlight the main findings of the results and there is no mention of the normative valuales (the other objective of the study).

  1. 236-251: I don't understand if the aim of the study was functional capacity and normative values, why the discussion is about obesity. Also the meaning of figure 1

8,9, 11.- (AR):  Indeed, we agree that the discussion section was poorly structured, and it included aspects that were not within the scope of the study. Thank you very much for your observations; we believe they are very accurate. As a result, we have completely restructured and rewritten the entire discussion. We have also removed references to the objectives, the graph, and the discussion on obesity and the diagnostic value of BMI.

  1. 229-235: What you are explaining here is a limitation of the study, as there is a problem of external validity to the general population. I recommend relocating and rephrase this to a limitations section.

10.- (AR): We have also reworked the paragraphs regarding the strengths, weaknesses, and utilities of this study, increasing the number of strengths and weaknesses.

  1. 256-347: The section mainly discusses aspects that were intended to be evaluated in the study. However, what is done mainly is to comment on them as in the results section without being a real discussion about why these results occur, the previous studies that support or contradict them, and the implications that the results have for the population and/or science.

12.(AR): We acknowledge your comment. We have addressed all the aspects you mentioned. In the discussion section, following your feedback, we have added the data extracted from published studies that focused on or included the older adult population, providing data in five-year age groups. Using these data, we calculated the annual percentage decline starting from 60 years of age (in some studies, it was from 65 years of age) and compared it with our own data. Furthermore, we have not found any published studies in PubMed journals conducted in Mexico with the Senior Fitness Tests battery specifically targeting the older adult population. Therefore, we are unable to compare normative data from other regions in Mexico. Additionally, we believe that comparing normative data among countries with different ethnicities, cultures, socioeconomic statuses, etc., may not provide substantial insights, particularly without delving deeply into explaining the differences observed in these studies, as such explanations would largely be speculative.

  1. 348-357: Here, table 7 and figure 2 are introduced for the first time, when they should have been introduced in the results section.

13.- (AR): We have included the previous explanation of Table 2 in the text. Additionally, we have moved Figure 2 (now Figure 1) to the results section.

14. 361-371: Translation: This is the section on strengths and weaknesses. I have already commented on other weaknesses throughout the review that should be added to this section. It would also be convenient to comment on the sample size (can normative values for the entire northwest of Mexico be established with 550 participants?).

14.- (AR): We have added these and other aspects to the strengths and weaknesses. Furthermore, we understand that the sample size of this study may lead one to think it is quite small, especially when compared to more robust studies with much larger samples often resulting from the aggregation of data from smaller studies conducted over an extended period by different authors. Acknowledging this aspect, we also want to highlight that many of these large-scale studies would not be possible without the foundation laid by previously published smaller studies. In Mexico, there are no published studies on functional capacity with large (or aggregated) samples, and we aspire to carry out such studies in the near future. This serves as an incentive that we believe can be derived from our study.

Regarding the sample size of our study, we also want to emphasize that it is not as small as it may appear. It is a representative sample of approximately one million older adults, which means that our sample comprises 5.5 out of every 1,000 older adults. This figure becomes even higher when considering the limitations we previously mentioned in the conclusions, such as focusing only on urban-dwelling older adults who are not institutionalized. While our study may be less robust than many others, it is no less valuable than other studies that have been published.

TABLES:

15.- In general, the foot of the tables is confused with the general text of the article. Please modify. Statistically significant differences are indicated in the tables with an asterisk. However, it is not commented anywhere which p-values will be considered significant or not. Additionally, a p-value of 0.049 is not the same as p<0.00001. Please indicate the p-values of the differences or at least put ranges. In general, the tables are not referenced in the text. Please correct this.

  1. (AR): Thank you for your observation. Indeed, it was quite confusing. We have modified the table captions and indicated *=p<0.05. It is true that the p-value can take different values, each with its own level of probability. It becomes more informative when accompanied by confidence intervals and effect sizes. However, considering the context of the article, where we are not studying the effect of an intervention but rather providing a descriptive and approximate comparison - not a detailed one - and considering the amount of data contained in the tables, we believe that adding the p-value, confidence interval, effect size, etc., would further complicate the readability of the tables. Honestly, we do not believe it would provide significant information in the context of this study, and it is also common in this type of work to express it this way.

16.-Table 7. The normative values expressed in 3 quartiles (q1, q2, q3) are not the best way to express these values. Usually, studies of this nature express them in 5 quartiles (for example, reference 23 with 10th, 25th, 50th, 75th, and 90th). Please justify the calculation methodologically in the METHODS section and consider expanding the quartiles for calculation (which I believe is vital to establish normative values).

16. (AR): Gracias por su observación. Hemos recalculado los valores de referencia, y los hemos expresado como se expresan de forma habitual en este tipo de artículos, como por ejemplo en la referencia 23, expresándolos ahora en la tabla 7, en forma de centiles 10th, 25th, 50th, 75th, and 90th.

17. FIGURES: Figures 1 and 2 appear in the discussion without being mentioned in either the methodology or results section. Please modify this. Figure 1 missed label of X side.

17. (AR): Thank you for your observation. We have recalculated the reference values and presented them in the customary manner for this type of articles, as exemplified in reference 23. These values are now expressed in Table 7 as 10th, 25th, 50th, 75th, and 90th percentiles (centiles).

18.- Please avoid the use of elderly

  1. (AR): We understand that there can be cultural interpretations of certain words across different countries. While "elderly" is commonly used in the medical context, we also acknowledge that it may carry derogatory connotations in some Anglo-Saxon countries. Therefore, we have replaced "elderly" with "old people" throughout the document, except in the titles of the 8 referenced publications where the word is specifically included.

Once again, we would like to express our gratitude for your time and the valuable observations you provided. We have addressed each point diligently, and your feedback has significantly improved our work. Thank you.

Round 2

Reviewer 2 Report

The authors have addressed all my requests and I believe that the article could be publishable.

Author Response

Thank you very much for your kind comments that have improved our work and for your kind response.

Reviewer 4 Report

Dear Authors,

 Thank you for your kind words and for expressing your gratitude. I appreciate your acknowledgment of the time and effort I invested in reviewing your manuscript.

 I'm glad to hear that you found my observations valuable and that they have contributed to improving your work. It is always rewarding to see authors diligently address the raised points and make significant enhancements based on reviewer feedback.

 Your willingness to incorporate the suggested revisions demonstrates your commitment to producing a high-quality research article. I commend your dedication and attention to detail.

After the modifications made, although my main concern (can normative values for the entire northwest of Mexico be established with 550 participants?) still remains, I believe your article deserves to be published.

Author Response

Again, thank you very much for your time, for your corrections and observations to our manuscript that have improved it remarkably. And thank you very much for your kind response.